# Monosaccharide Composition and In Vitro Activity to HCT-116 Cells of Purslane Polysaccharides after a Covalent Chemical Selenylation

**DOI:** 10.3390/foods11233748

**Published:** 2022-11-22

**Authors:** Ling-Yu Li, Qing-Yun Guan, Ya-Ru Lin, Jun-Ren Zhao, Li Wang, Qiang Zhang, Hong-Fang Liu, Xin-Huai Zhao

**Affiliations:** 1School of Biology and Food Engineering, Guangdong University of Petrochemical Technology, Maoming 525000, China; 2Key Laboratory of Dairy Science, Ministry of Education, Northeast Agricultural University, Harbin 150030, China; 3Research Centre of Food Nutrition and Human Healthcare, Guangdong University of Petrochemical Technology, Maoming 525000, China; 4Harbin Comprehensive Inspection and Detection Centre for Product Quality, Harbin 150036, China

**Keywords:** purslane, polysaccharide, selenylation, monosaccharide composition, HCT-116 cells, nutraceutical effect

## Abstract

The anti-cancer effects of selenylated plant polysaccharides are a focus of research. As a natural plant with extensive biological effects, there have been few studies related to edible purslane (*Portulaca oleracea* L.). Thus, in this study, soluble *P. oleracea* polysaccharides (PPS) were extracted from the dried *P. oleracea* and then selenylated chemically using the HNO_3_-Na_2_SeO_3_ method to obtain two selenylated products, namely, SePPS1 and SePPS2. Compared with the extracted PPS, SePPS1 and SePPS2 had much higher Se contents (840.3 and 1770.5 versus 66.0 mg/kg) while also showing lower contents in three saccharides—arabinose, fucose, and ribose—and higher contents in seven saccharides including galactose, glucose, fructose, mannose, rhamnose, galacturonic acid, and glucuronic acid, but a stable xylose content demonstrated that the performed chemical selenylation of PPS led to changes in monosaccharide composition. Moreover, SePPS1 and SePPS2 shared similar features with respect to monosaccharide composition and possessed higher bioactivity than PPS in human colon cancer HCT-116 cells. Generally, SePPS1 and SePPS2 were more active than PPS with respect to cell growth inhibition, the alteration of cell morphology, disruption of mitochondrial membrane potential, intracellular reactive oxygen species (ROS) generation, the induction of cell apoptosis, and upregulation or downregulation of five apoptosis-related genes and proteins such as Bax, Bcl-2, caspases-3/-9, and cytochrome C, that cause cell apoptosis and growth suppression via the ROS-mediated mitochondrial pathway. SePPS2 consistently showed the highest capacity to exert these observed effects on the targeted cells, suggesting that the performed chemical selenylation of PPS (in particular when higher degrees of selenylation are reached) resulted in an increase in activity in the cells. It can thus be concluded that the performed selenylation of PPS was able to incorporate inorganic Se into the final PPS products, changing their monosaccharide composition and endowing them with enhanced nutraceutical and anti-cancer effects in the colon.

## 1. Introduction

Usually, non-digestible oligosaccharides, non-starch polysaccharides, and resistant starch are well-regarded, and they are important components of dietary fibers [1]. As a result of non-starch polysaccharides not being able to be decomposed by the digestive enzymes in the digestive tract, they have important physiological functions in the body. It is known that dietary fibers have several nutraceutical effects, as they are able to reduce the risks of diseases such as pre-diabetes, coronary heart disease, obesity, type 2 diabetes, cardiovascular disease, and some cancers [2]. Therefore, non-starch polysaccharides from various food sources have been investigated with respect to their nutraceutical potential. However, non-starch polysaccharides from various non-conventional food sources have still been insufficiently investigated. Purslane (*Portulaca Oleracea* L.) is a non-conventional but edible annual herb. *P. oleracea* has both soluble and non-starch polysaccharide components (i.e., *P. oleracea* polysaccharides, PPS), which are regarded as the main bioactive substances in *P. oleracea*. Meanwhile, PPS is composed of monosaccharide units such as glucose, mannose, arabinose, galactose, and galacturonic acid [3]. PPS and other natural polysaccharides have attracted extensive attention due to possessing physiological activities which might be beneficial to the human body. Previous results have shown that polysaccharides may possess anti-tumor [4], anti-oxidation [5], immuno-regulation [6], and other nutraceutical effects. For example, the polysaccharides from *Ganoderma lucidum* have been shown to exert an inhibitory effect on the growth of hepatoma and Lewis lung cancer cells [7,8], while those from *Lentinus edodes* have also demonstrated the growth inhibition of colon cancer HT-29 cells [9]. In addition, the polysaccharides isolated from mango pomace have been reported to inhibit the proliferation of human colon cancer HCT-116 cells by inducing apoptosis [10]. With respect to PPS, they have been reported to exert an immune-regulatory effect on thymic lymphocytes in Wistar rats because of their ability to enhance the proliferation of the spleen, as well as thymocyte T and B lymphocytes [6]. More importantly, PPS are also regarded as possessing anti-tumor activity because in cervical cancer cells, they have demonstrated the ability to inhibit the proliferation of cervical cancer cells, trigger DNA damage, and induce cell apoptosis [11]. However, to the best of our knowledge, whether the targeted chemical modification of PPS (e.g., through selenylation) might result in a change in activity has been less thoroughly investigated. Thus, the in vitro bioactivities of selenylated PPS (i.e., SePPS) must be assessed and compared with those of the unmodified PPS.

It is known that structural changes in polysaccharides can affect their properties. Thus, natural polysaccharides have been modified chemically by means of various approaches, with the aim of altering their physicochemical properties and, in particular, their nutraceutical effects. It has been found that chemical acetylation can improve the emulsification performance of millet polysaccharides, while pumpkin polysaccharides after chemical acetylation exhibit a higher anti-oxidant capacity for scavenging 1,1-diphenyl-2-picrylhydrazyl, superoxide anion, and other free radicals [12,13]. Additionally, the carboxymethylated polysaccharides from *Poria cocos* have been proven to exert a stronger anti-proliferation effect on both HT-29 and SGC-7901 cells [14]. Chemical selenylation can also be used to modify the properties of natural polysaccharides. Se is well known as an important and essential minor element and has many beneficial functions in the body. Generally speaking, the best-studied bioactivities of Se include its anti-oxidation and anti-tumor effects, insulin simulation, and immuno-modulation. Chemical selenylation can, in theory, be used to incorporate inorganic Se into the functional groups (i.e., −OH) of macromolecules such as polysaccharides and proteins [15], resulting in the formation of selenylated polysaccharides and proteins, respectively. Selenylated polysaccharides are regarded as a kind of organic Se compound, potentially possessing bioactivities originating from both Se and the polysaccharides. Previous studies have demonstrated that selenylated polysaccharides have a higher activity than either Se or polysaccharides alone [16]. Investigating whether the chemical selenylation of PPS could result in enhanced activity in vitro is of great significance.

Therefore, we hypothesized that the chemical selenylation of PPS would have the potential to alter the saccharide composition of PPS, as well as change the nutraceutical effect on human colon cancer HCT-116 cells, as a result of the covalent incorporation of Se into PPS molecules. In this study, the soluble PPS were thus extracted from *P. oleracea* using water and were then modified by a reaction system composed of Na_2_SeO_3_ and HNO_3_ to prepare two selenylated polysaccharides, namely SePPS1 and SePPS2, with different contents of Se (or selenylation extents). Afterward, PPS, SePPS1, and SePPS2 were evaluated and compared for their monosaccharide composition and nutraceutical effect on the HCT-116 cells. Five indices, including growth inhibition, cell morphology, reactive oxygen species (ROS) generation, mitochondrial membrane potential (MMP) loss, and apoptosis induction, were assessed to reflect the targeted nutraceutical effect, while several important genes and proteins in the cells were evaluated for relative expression changes to support the in vitro activities of SePPS1 and SePPS2 towards the cells. This study aimed to reveal whether the monosaccharide composition and nutraceutical effect (to be more specific, anti-cancer effect) of PPS could be impacted by the performed chemical selenylation as well as the obtained selenylation extent.

## 2. Materials and Methods

### 2.1. Materials and Reagents

Fresh purslane, planted in Rizhao city (Shandong Province, China), was collected in July 2020 and identified by Prof. Dai-Di Che from Northeast Agricultural University (Harbin, China). The 3-(4,5-dimethyl-2-thiazolyl)-2,5-diphenyltetrazole, McCoy’s 5A medium, and ammonium bromide (MTT) were provided by the Sigma Chemicals Co. (St. Louis, MO, USA). Fetal bovine serum (FBS) was purchased from Thermo Fisher Scientific Inc. (Cleveland, OH, USA), while 5-fururacil (5-FU) was purchased from Aladdin Company (Shanghai, China). Ultrapure water generated from Milli-Q Plus, a product of Millipore Co. (New York, NY, USA) and was used in this study. Other chemicals used in this study were of analytical grade.

The ROS assay kit, Annexin V-FITC/PI kit, Hoechst 33258 kit, and mitochondrial membrane potential assay kit (JC-1) were bought from the Beyotime Biotechnology Institute (Beijing, China). The used antibodies, including Bax, Bcl-2, caspases-3/-9, cytochrome C, β-actin, horseradish peroxidase-labeled goat anti-mouse IgG, and goat anti-rabbit IgG, were purchased from Cell Signaling Technology (Danvers, MA, USA).

### 2.2. Cell Line and Cell Culture

HCT-116 cells, isolated from the colon of an adult male with colon cancer, were purchased from the Cell Bank of Shanghai Institute of Biochemistry and Cell Biology (Shanghai, China). As required, the cells were cultured in McCoy’s 5A medium supplemented with 10% FBS in a humidified incubator with 5% CO_2_ at 37 °C. The cells were digested with 0.25% trypsin digestion solution for 1 min every 2–3 d. All cell experiments were performed using the cells in the logarithmic growth phase.

### 2.3. Preparation and Chemical Selenylation of P. oleracea Polysaccharides

According to the reported method [17], fresh *P. oleracea* was washed after removing its roots and then dried in the shade. The dried *P. oleracea* (including the leaves, flowers, and stalks) was crushed into powder and mixed with distilled water at a fixed solid-liquid ratio (1:20, *w/v*). After that, a thermostable α-amylase (20 U/mL) was added and kept at 90 °C for 4 h to degrade the starch substances. After centrifugation at 8000× *g* for 15 min, an alkaline protease alcalase (100 U/mL) was added to the collected supernatant, kept at 55 °C for 8 h, and centrifuged again at 8000× *g* for 15 min. The yielded supernatant was concentrated to one-tenth of the original volume, mixed with absolute ethanol of a three-fold volume, and then incubated overnight at 4 °C. The obtained precipitates (i.e., PPS) were then washed with absolute ethanol three times, soaked in absolute ethyl ether to remove the fatty substances, dialyzed to remove the salts and small molecule impurities, and finally freeze-dried. The obtained PPS were stored at −20 °C before use.

The selenylated PPS were prepared, as previously described [18], by employing a selenylation system composed of Na_2_SeO_3_ and HNO_3_. In brief, PPS powder (300 mg) was dispersed in 5% HNO_3_ of 20 mL, mixed with 30 or 45 mg Na_2_SeO_3_, held at 75 °C for 8 h, and then cooled to 20 °C. The absolute ethanol of a three-fold volume was added to the selenylation system, and incubated at 4 °C overnight. The precipitates were further collected, washed with anhydrous ethanol five times, and freeze-dried to obtain two selenylated PPS products, namely SePPS1 and SePPS2, respectively. Additionally, an equal amount of PPS was mixed with Na_2_SeO_3_ in the absence of HNO_3_ to perform the same treatments, and aiming to obtain the control PPS.

### 2.4. Determination of Ash, Total Saccharide and Se Contents

Ash content was measured using the method of AOAC 937.09 [19]. Total saccharide was detected by the phenol-H_2_SO_4_ as previously described [20]. Moreover, the Se content was measured by an inductively coupled plasma-mass spectrometer (Agilent Technologies, Santa Clara, CA, USA) according to a previous study [21].

### 2.5. Determination of Monosaccharide Composition

Monosaccharide composition was detected using the reported conditions [22]. Briefly, the samples of 5 mg were dissolved in 2 mL of trifluoroacetic acid (4 mol/L) and hydrolyzed at 110 °C for 4 h. After cooling to 20 °C, the obtained hydrolysates were dried using a nitrogen blow, added with 1 mL of pyridine, 0.3 mL of hexamethyldisilane, and 0.6 mL of trimethylchlorosilane immediately before being kept at 50 °C for 40 min. Afterward, the supernatants were obtained by centrifugation at 8000× *g* for 5 min and detected for monosaccharide composition using a GC-MS analyzer (Type 7000D, Agilent Technologies, Santa Clara, CA, USA) coupled with a DB-5MS column (30 m × 0.25 mm × 0.25 μm).

### 2.6. Assay of Growth Inhibition and Colony Formation

The effects of the samples on cell viability were evaluated, as previously described [23], to reflect their growth inhibition on HCT-116 cells. In detail, HCT-116 cell solutions (1 × 10^5^ cells/mL, 100 μL) were plated into 96-well plates and incubated at 37 °C for 24 h to ensure cell adherence. After discarding the supernatants, 100 μL 5-FU (100 μmol/L, positive group) or the three polysaccharide samples at various dose levels (50–800 μg/mL) were added to treat the cells for 24 or 48 h. Afterward, the medium was discarded, while an MTT solution of 100 μL was added to each well. The cells were incubated at 37 °C for 4 h. The supernatants were also aspirated, while dimethyl sulfoxide (DMSO) of 100 μL was added into each well. A microplate reader (Bio-Rad Laboratories, Hercules, CA, USA) was then used to measure the optical density (OD) values at 450 nm, which were used to calculate the percentages of growth inhibition. As usual, the control cells without 5-FU or a sample treatment were set with 100% cell viability.

The long-term inhibitory effect of the samples on the cells was measured by a colony formation assay, as previously described [24]. In brief, the cells of 2 mL (1 × 10^3^ cells/mL) were seeded into 6-well plates and cultured in the medium with 5% FBS for 24 h. After medium discarding, 2 mL of the samples at the dose levels of 400 and 800 μg/mL were added into each well while the cells were incubated for 24 h again. The medium was changed every 3 d till the cells were cultured for 14 d. The cells were then fixed with polymethanol for 20 min and stained with crystal violet for 5 min. The plates were photographed after drying under a microscope (Olympus, Tokyo, Japan) to collect their images.

### 2.7. Assay of Cell Apoptosis by Flow Cytometry

The proportion of apoptotic cells was determined using flow cytometry and Annexin V-FITC/PI double staining as previously described [25]. The cells of 2 mL were seeded into 6-well plates (3 × 10^4^ cells/well) and cultured for 24 h. The medium was removed, while the samples (dose levels of 400 and 800 μg/mL) of 2 mL were added into each well to treat the cells for 24 and 48 h. The cells were washed with PBS and centrifuged at 1000× *g* for 5 min. After discarding the supernatants, the cells were added with Annexin V-FITC binding buffer of 195 μL and Annexin V-FITC of 5 μL, kept at 20 °C for 15 min, and then resuspended with 10 μL PI. After passing a sieve of 300 mesh, the cells were detected by flow cytometry (Type BDFACS Aria II, BD Bioscience, Franklin Lakes, NJ, USA) to determine the proportion of the total apoptotic cells.

### 2.8. Hoechst 33258 Staining

A fluorescent probe Hoechst 33258, was used for nuclear staining as previously described [26]. The cells of 2 mL were seeded into 6-well plates (1 × 10^6^ cells/well) and cultured for 24 h. After medium removal, 2 mL of the samples at the dose levels of 400 and 800 μg/mL were added into each well to treat the cells for 24 and 48 h. After discarding the medium, the cells were treated with a fixing solution of 500 μL for 10 min. The cells were washed with PBS, with added Hoechst 33258 dye solution of 500 μL at 20 °C for 5 min, before being washed with PBS again. The cells were then imaged using a microscope (Zeiss Axio Observer A1m, Carl Zeiss, Oberkochen, BW, Germany), with an objective of 20-folds and 350/460 nm excitation and emission wavelengths.

### 2.9. Assay of Intracellular ROS

The level of intracellular ROS was detected as previously described [27]. The cells of 2 mL were seeded into 6-well plates (1 × 10^6^ cells/well) and cultured for 24 h. After medium discarding, 2 mL of the samples (dose levels of 400 and 800 μg/mL) were added to each well to treat the cells for 24 and 48 h, respectively. The culture medium was collected, while the cells were washed with PBS three times. The cells were then mixed with the collected culture medium and washed with PBS again. Afterward, the cells were added with the diluted DCFH-DA of 1 mL, incubated at 37 °C for 30 min, washed with serum-free medium, centrifuged at 8000× *g* for 5 min to discard the supernatants, and finally re-suspended with PBS of 1 mL. The value of the fluorescence intensity for each well was measured using the respective 488/525 nm of excitation and emission wavelengths at a fluorescent microplate reader (Infinite 200, Tecan, Mannedorf, Switzerland). The relative ROS level of the treated cells was expressed as a percentage of the control cells that were regarded with a relative ROS level of 100%.

### 2.10. Assay of Mitochondrial Membrane Potential Loss

The JC-1 fluorescent probe was used to measure the changes in mitochondrial membrane potential (MMP) in the cells as previously described [28]. The cells of 2 mL were seeded into 6-well plates (1 × 10^5^ cells/well) and cultured for 24 h. After medium discarding, 2 mL of the samples (dose levels of 400 and 800 μg/mL) were added to each well to treat the cells for 24 and 48 h, respectively. The cells were washed with PBS, added with the medium, and a JC-1 staining solution of 1 mL before being incubated at 37 °C for 20 min. The cells were washed with 1 × JC-1 staining buffer, centrifuged at 1000× *g* for 5 min to discard the supernatants, and then re-suspended with PBS of 1 mL. Afterward, the fluorescence intensity was measured using the respective 490/530 nm of excitation and emission wavelengths at the fluorescent microplate reader. The MMP loss of the cells was reflected by using the ratio of red/green fluorescence intensity.

### 2.11. Assay of Gene Expression Using RT-PCR Assay

The cells of 2 mL were seeded into 6-well plates (1 × 10^5^ cells/well) and cultured for 24 and 48 h. After medium removal, the samples (dose levels of 400 and 800 μg/mL) of 2 mL were used to treat the cells for 24 h. Genomic DNA was removed, while the total RNA was extracted according to the suggested procedures of the Tiangen RNAprep Pure Cell Total RNA Extraction Kit (Tiangen Company, Beijing, China).

RT-PCR assay was performed using the FastKing One-Step RT-qPCR Kit and the Applied Biosystems StepOnePlus Real-time PCR System (Life Technologies Corp., Carlsbad, CA, USA), while β-actin was used as an internal reference gene. The primer sequences of β-actin, Bax, Bcl-2, caspases-3/-9, and cytochrome C were designed by the Sangon Biotech Co. (Shanghai, China) (Table 1). PCR data were analyzed and calculated by the 2^−∆∆Ct^ method as previously described [29]. The expression levels of these genes in the control cells without sample treatment were fixed at 1.00 fold.

### 2.12. Western-Blotting Analysis

The cells of 2 mL were cultured in the culture flasks with a capacity of 25 cm^2^ (2 × 10^5^ cells/flask) for 24 h. After medium discarding, the samples (dose level of 800 μg/mL) of 4 mL were added to treat the cells for 24 h. The cells were washed with PBS at 4 °C three times and lysed with the RIPA cleavage buffer and 1 mmol/L PMSF (Beyotime, Shanghai, China) on ice for 30 min. The lysed cells were centrifuged at 12,000× *g* for 5 min at 4 °C to extract total protein, while the total protein concentration was determined using the BCA protein analysis kit (Beyotime, Shanghai, China).

The obtained protein samples (50 μg) were added with the loading buffer (5×) in a ratio of 4:1 (*v*/*v*), boiled for 5 min, separated by SDS-PAGE (12%), and then blotted onto the nitrocellulose membrane. The membrane was placed in 5% skim milk and sealed at 37 °C for 2 h before being added with the primary antibody (1:1000 dilution) and incubated overnight at 4 °C. The membrane was washed with the TBST buffer three times and incubated with the horseradish peroxidase-labeled goat anti-mouse IgG and goat anti-rabbit IgG antibody (1:5000 dilution) at 20 °C for 2 h. Afterward, the enhanced chemiluminescence (ECL) and chemiluminescence imaging system (Bio-Rad Laboratories, Hercules, CA, USA) were used to detect the protein bands. The Image J software (National Institutes of Health, Bethesda, MD, USA) was used to analyze the gray levels of the targeted protein bands and normalized them to the level of β-actin as previously described [30].

### 2.13. Statistical Analysis

All reported data were collected from three experiments or analyses and expressed as the means ± standard deviations in this study. The differences between the mean values were determined by one-way ANOVA variance analysis and Duncan’s multiple range test (*p* < 0.05) using the SPSS version 16.0 software (SPSS, Inc., Chicago, IL, USA).

## 3. Results

### 3.1. Several Chemical Features of the Prepared Polysaccharide Samples

The total contents of saccharide and ash in PPS, measured by the phenol-H_2_SO_4_ and ignition gravimetric methods, were 861.6 and 85.5 g/kg, respectively. After the chemical selenylation, both SePPS1 and SePPS2 obtained a Se incorporation of two extents based on the fact that they contained much higher Se contents. Specifically, the content of Se in the unselenylated PPS was 66.0 mg/kg (dry basis), whilst the contents of Se in SePPS1 and SePPS2 were up to 840.3 and 1770.5 mg/kg (dry basis), respectively. Compared with PPS, both SePPS1 and SePPS2 had increased Se contents by respective 8- and 26-folds, demonstrating the successful selenylation of PPS by the employed HNO_3_-Na_2_SeO_3_ system. As the measured Se contents of PPS, SePPS1 and SePPS2 were different, this study thus could reveal whether the performed chemical selenylation, as well as the resultant selenylation extent, exerted an impact on the bioactivities of SePPS1 and SePPS2 in the targeted cells.

The results of the GC-MS analysis indicated that PPS, SePPS1, and SePPS2 were mainly composed of eleven monosaccharide units, as the results are shown in Figure 1 and Table 2. In detail, eleven monosaccharide units, including xylose, fucose, mannose, fructose, glucuronic acid, galacturonic acid, rhamnose, ribose, galactose, arabinose, and glucose, were detected in PPS, with the relative percentages of 1.05, 2.16, 2.46, 2.46, 2.88, 7.56, 9.06, 10.13, 12.33, 12.69, and 21.92, respectively. Due to the technical limitation of our research group in saccharide identification, some minor monosaccharides in the PPS were still unidentified. Interestingly, the monosaccharide composition of SePPS1 and SePPS2 was different from that of PPS. To be more specific, SePPS1 and SePPS2 had lower levels of arabinose, fucose, and ribose than PPS and subsequently showed higher levels in the other seven monosaccharides than PPS. However, it was found that the content of xylose was stable in the three samples. Additionally, the data comparison suggested that SePPS1 and SePPS2 might be regarded to have similar monosaccharide compositions (Table 2). Why the conducted chemical selenylation led to the loss of arabinose, fucose, and ribose in SePPS1 and SePPS2 was unknown to us and, thus, is not discussed here, which might be investigated in a future study. Overall, the used chemical selenylation altered the monosaccharide composition of SePPS1 and SePPS2 and incorporated Se into the polysaccharide molecules. Consequentially, it was inferred that SePPS1 and SePPS2 might exhibit a different activity, compared with PPS in the cells, because they possessed different monosaccharide compositions and Se contents in comparison with PPS.

### 3.2. The Growth Inhibitory Effect of the Polysaccharide Samples on HCT-116 Cells

The growth inhibition of HCT-116 cells in response to three polysaccharides at the dose levels of 50–800 μg/mL are shown in Figure 2. PPS, SePPS1, and SePPS2 all showed similar functions to the positive control 5-FU to inhibit the growth of HCT-116 cells, demonstrating their activities to the cells. With a cell treatment of 24 h, the values of the growth inhibition for PPS, SePPS1, and SePPS2 were 14.2–24.4%, 15.2–29.8%, and 17.2–43.4%, respectively. If the cells were treated with the samples for 48 h, the resultant values of the growth inhibition were increased to 19.3–37.0%, 22.1–41.2%, and 25.2–49.4%, respectively. All data demonstrated the obvious growth suppression of the assessed polysaccharide samples in the cells. Consistently, the longer cell treatment time and higher sample dose caused a greater inhibition, suggesting the time- and dose-dependent growth suppression on the cells. Meanwhile, PPS and SePPS2 under the same conditions caused the lowest and highest inhibition, respectively, declaring that the used selenylation and higher selenylation extent brought about a higher potential to inhibit cell growth.

The Hoechst 33258 staining results (Figure 3) also confirmed the activities of the assessed samples in the cell. In brief, compared with the negative control cells, the cells from other groups showed reduced cell volume, concentrated chromatin, split nuclei into fragments to form apoptotic bodies, and typical apoptotic morphology, reflecting the classic DNA damage and cell apoptosis. Moreover, the results from the colony formation assay (Figure 4) also proved that the samples had a long-term inhibitory effect on the cells. Furthermore, PPS and SePPS2 under the same conditions exhibited the lowest and highest long-term inhibitory effect on the cells, respectively, suggesting again that the used selenylation and the higher selenylation extent induced a greater long-term inhibitory effect for the modified PPS.

### 3.3. Effects of the Polysaccharide Samples on MMP Loss and ROS Formation

MMP changes, in the cells in response to the exposure of the three samples, reflected as changed ratio of red/green fluorescence intensity, are shown in Figure 5a. Compared with the control cells (i.e., without sample treatment), the cells treated by the samples showed MMP loss because the detected ratios of red/green fluorescence intensity were reduced. In the control cells, the ratio of red/green fluorescence intensity was 9.00 (24 h) or 9.91 (48 h). When the cells were treated for 24 h by PPS, SePPS1, and SePPS2, the respective ratios of red/green fluorescence intensity decreased to 7.59, 5.65, and 2.97. If the cells were treated for 48 h by the samples, the corresponding ratios of red/green fluorescence intensities decreased to 8.30, 5.89, and 3.25. The samples showed an ability to decrease the ratios of red/green fluorescence intensity (the indicator of membrane permeability), while the higher dose level of samples led to a remarkable reduction for this indicator. Furthermore, PPS and SePPS2 under the same conditions resulted in the highest and lowest ratios of red/green fluorescence intensity in the cells, declaring their highest and lowest ability to cause MMP loss. This fact meant that the used chemical selenylation and higher selenylation extent enhanced the ability of the selenylated PPS products to damage MMP.

When the samples were used to treat the cells for 24 and 48 h, the ROS levels of the treated cells also showed an obvious increase (Figure 5b). When the cells were treated for 24 h by PPS, SePPS1, and SePPS2, the measured ROS levels were 117.0%, 124.0–139.0%, and 147.0–169.7%, respectively. If the cells were treated by the samples for 48 h, the resultant ROS levels increased up to 130.3%, 147.7–160.3%, and 188.0–226.7%, respectively. Compared with the control cells, the treated cells had about a 20–130% increase in ROS formation. Obviously, SePPS2 showed the highest ability to promote ROS formation, while PPS exhibited the lowest ability to elevate ROS generation. In addition, the higher dose level also resulted in more ROS generation in the cells. All these results suggested that the used chemical selenylation and higher selenylation extent endowed the selenylated PPS products with a higher capacity to promote ROS formation in the cells. Considering the well-known fact that MMP loss and unusual ROS levels in the cells might induce cell apoptosis, SePPS1 and SePPS2 should be further assessed for their apoptosis induction in the cells.

### 3.4. Apoptosis Induction of the Polysaccharide Samples towards HCT-116 Cells

When the cells were treated by the samples for 24 or 48 h, their total apoptotic cells (Q2 + Q4) were evaluated using flow cytometry and the Annexin V-FICT/PI double staining (Figure 6). Subsequently, the control cells were found with the lowest number of total apoptotic cells (11.2%, 24 h; 9.5%, 48 h) (Table 3). When the cells were treated for 24 h by PPS, SePPS1, and SePPS2, the respective numbers of the total apoptotic cells were 14.8%, 16.8–17.2%, and 18.1–20.5% (Table 3). If the cells were treated for 48 h by the samples, the number of total apoptotic cells was enhanced to 20.2%, 23.3–24.3%, and 31.9–35.9%, respectively (Table 3). The samples were, thus, considered to have the ability to induce cell apoptosis. In addition, PPS and SePPS2 under the same conditions brought about the lowest and highest numbers of total apoptotic cells, suggesting that the applied chemical polysaccharide selenylation and higher selenylation extent consistently led to the higher apoptosis induction for the selenylated PPS products. Furthermore, it was also seen that the apoptosis induction of these samples was consistent with their growth inhibition and their potential to cause MMP loss and ROS formation.

### 3.5. Effects of the Polysaccharide Samples on Gene and Protein Expression

The RT-PCR assay results (Table 4) demonstrated that the samples in the cells could regulate the expression of the targeted five genes at the mRNA level. When the cells were treated for 24 h by PPS, SePPS1, and SePPS2, the respective expression levels of Bax, Bcl-2, caspase-3, caspase-9, and cytochrome C were 1.28–2.3, 0.61–0.96, 1.37–2.39, 1.06–2.40, and 1.04–1.64 folds. The samples thus up-regulated or down-regulated these apoptosis-related genes. In total, SePPS2 and PPS showed the highest and lowest ability to regulate these genes, respectively. Thus, the used chemical selenylation and higher selenylation extent endowed the selenylated PPS products with a higher potential to regulate the expression of the five genes.

The samples were also capable of up-regulating or down-regulating the expression of the five apoptosis-related proteins (Figure 7). Compared with the control cells, the PPS-treated cells at 24 h showed the up-regulated protein expression for Bax, caspase-3, caspase-9, and cytochrome C (3.29-, 1.34-, 1.10-, and 1.98-folds), while the SePPS1-treated cells at 24 h also had up-regulated expression levels for the four proteins (3.34-, 1.35-, 1.15-, and 1.99-folds). In addition, the SePPS2-treated cells at 24 h were detected with much-up-regulated expression levels of 3.58-, 1.45-, 1.26-, and 2.13-folds for the four proteins. Meanwhile, PPS, SePPS1, and SePPS2 caused the Bcl-2 down-expression of 0.99-, 0.56-, and 0.32-folds, respectively. The western-blotting results thus proved that the samples had different capacities to regulate the expression of the five apoptosis-related proteins, and the used chemical selenylation and higher selenylation extent ensured that the selenylated PPS product had a higher activity to regulate the critical protein expression. Considering the results reflected in Figure 5 and Figure 7, it could be inferred briefly that SePPS1 and SePPS2 obtained higher activity against HCT-116 cells to cause growth inhibition and cell apoptosis via the ROS-mediated mitochondrial pathway.

## 4. Discussion

In general, polysaccharides can be served as a stabilizer to maintain the physical stability of food systems such as beverages or to improve the structural and rheological properties of the applied foods [31,32]. Polysaccharides also have many beneficial functions. For example, *Salvia miltiorrhiza* polysaccharides could protect H9c2 cells from the H_2_O_2_-induced apoptosis and myocardial function [33], while the polysaccharides of Coix seed could modulate gut microbial compositions, activate the IGF1/PI3K/AKT signaling pathways, and thereby exhibit a hypoglycemic effect [34]. The polysaccharide of marine bivalves was also verified to have an anti-proliferative effect on several cancer cell lines, including breast (MDA-MB-231), cervical (Hela), liver (HepG2), and colon (HT-29) cancer cells [35]. The chemical modification of natural polysaccharides is achieved by incorporating various chemical groups into polysaccharide molecules. The chemical selenylation of polysaccharides means the covalent incorporation of H_2_SeO_3_ molecules into saccharide units and thus ensures altered bioactivity. It was reported that the selenylated polysaccharides of sweet potato could inhibit tumor growth at a dose level of 1274 mg/kg in both time- and dose-dependent manner [36], while those from tea at 1520 mg/kg were able to inhibit cell proliferation and induce cell apoptosis in human breast cancer MCF-7 cells [37]. In addition, the selenylated polysaccharides from ginseng could induce the apoptosis of HL-60 cells [38]. These non-starch polysaccharides are non-digestible in the intestine and thus can pass through the intestine to exert their effects on colon cancers. HCT-116 cells have been used as a typical cell model to evaluate the bioactivities of the targeted substances. It was revealed that the anti-tumor effect of lignans isolated from *Combretum fruticosum* on HCT-116 cells was mainly due to their growth inhibition [39], while the pepsin-treated casein fraction exhibited anti-cancer potential on HCT-116 cells via causing the death of apoptotic cells [40]. Inconsistent with these reported results, the present study also showed that PPS had an anti-cancer potential via growth inhibition and apoptosis induction. More importantly, referring to these evaluation indicators, the present study also demonstrated that this polysaccharide selenylation was sufficient to enhance the nutraceutical effect of PPS on the cells, as the selenylated PPS products (especially that with a higher selenylation extent) were detected with enhanced growth inhibition and apoptosis induction. Moreover, it is worth noting that cultivation of *P. oleracea* or other edible plants in the Se-rich soil might cause a higher polysaccharide selenylation and obtain a higher nutraceutical effect for the resultant plant polysaccharides. This speculation might be useful to guide agricultural production for healthy foodstuffs in the future.

Apoptosis is a programmed process of cell death in which the cells actively move toward the end of life in accordance with certain procedures. Thus, apoptosis induction is one of the important indices in evaluating potential anti-cancer activities of the targeted substances in cancer cells. The Se-containing polysaccharides from *Artemisia sphaerta* exhibited a stronger anti-proliferative activity against HepG-2, A549s, and H1975 cells by inducing cell apoptosis and cell cycle arrest [41], while the fire-thorn selenylated polysaccharides could inhibit the growth of triple-negative breast cancer cells through inducing cell apoptosis [42]. ROS are the natural products of cellular aerobic metabolism; however, higher intracellular ROS levels can lead to irreversible oxidative damage on cells and then induce cell apoptosis. Moreover, the periodical opening of the mitochondrial permeability transition porin (MPTP) in cells can prevent the excessive accumulation of a positive charge in the inner and outer membranes. When the cells are stimulated, the continuous non-specific opening of MPTP leads to the increased osmotic pressure of the mitochondrial matrix, and the release of these pro-apoptotic factors, such as cytochrome C, AIF, and pro-caspases from the mitochondria, subsequently triggers cell apoptosis [43,44]. Thus, MMP loss, ROS generation, and the regulated expression of the apoptosis-related genes or proteins are important indices to confirm the anti-tumor activities of the targeted substances. In this study, the assessed samples showed a clear ability to cause MMP loss and promote ROS formation in the cells and were detected with apoptosis induction by enhancing the total apoptotic cell numbers as well as the regulation of these apoptosis-related genes and proteins. Thus, this study shared a conclusion consistent with these aforementioned studies.

It is well known that the aspartate-specific cysteinyl proteinases (i.e., caspases) are closely related to the apoptosis of eukaryotic cells and play a key role in multiple apoptosis pathways. Under normal conditions, the caspase enzymes exist in the form of inactive zymogens. When the cells are stimulated by apoptosis signals, the caspase upstream of the apoptosis pathway can activate the caspase downstream, while a cascade reaction occurs to transmit the apoptosis signals to the apoptotic substrates. The apoptosis initiators, such as caspases-2/-8/-9/-10, are upstream of the cascade reaction, while the apoptotic executors, including caspases-3/-6/-7, are downstream of the cascade reaction. These apoptosis executors are activated by the upstream promoters and then interact with specific substrates, which will cause changes in the cell morphology, as well as the properties of cells, and eventually lead to cell apoptosis. The first molecule identified as an apoptosis executor was cytochrome C. Moreover, the Bcl-2 family is composed of a series of pro-apoptotic members, including Bax, Bak, Bad, Bcl-x, Bid, Bik, Bim, and Hrk, and several anti-apoptotic members such as Bcl-2, Bcl-xL, Bcl-w, Bfl-1, and Mcl-1 [45]. When cells are stimulated to produce stress, some pro-apoptotic factors in the Bcl-2 family, such as Bax, will be activated. After activation, the Bax will cause the formation of permeable pores on the surface of the mitochondria and release pro-apoptotic factors such as cytochrome C. Afterward, cytochrome C binds to the carboxyl end of the apoptosis activating factor (apoptotic protein activating factor-1, Apaf-1), while the inactivated precursor protein pro-caspase-9 binds with the amino end of the other end to form apoptotic bodies, which leads to the shearing and activation of caspase-9. When caspase-9 is sheared and activated, other caspases, such as caspase-3, will be activated, resulting in the cascade reaction and cell apoptosis. Using these critical indicators, natural and modified substances, including polyphenols and polysaccharides, have been assessed for their potential to induce cell apoptosis via the mitochondrial pathway. For example, it was reported that the selenylated tea polysaccharides induced cell apoptosis through the mitochondrial pathway, as well as intracellular ROS production, increasing the Bax/Bcl-2 ratio and caspases-3/-9 activation [37], while the selenylated polysaccharides of ginseng also caused cell apoptosis in human promyelocytic leukemia (HL-60) cells via the mitochondrial-mediated pathway [38]. A phenolic extract from oleaster (*Olea europaeavar*. sylvestris) leaves was also suggested to induce caspase-dependent apoptosis in HCT-116 cells via the mitochondrial pathway [46]. In addition, the scutellarein from *Scutellaria barbata* could induce the apoptosis of HCT-116 cells through the ROS-mediated mitochondria-dependent pathway [47], while the polysaccharides from *Brassica juncea* induced the apoptosis of the colorectal cancer cells via both mitochondrial- and caspase-dependent pathways [48]. In this study, the polysaccharide samples were also verified to have the ability to enhance ROS formation and to regulate these apoptosis-related proteins (Bax, Bcl-2, cytochrome C, and caspases-3/-9), which were regarded to induce the ROS-mediated mitochondria-dependent apoptosis. Thereby, this study shared a conclusion consistent with these previous studies.

## 5. Conclusions

The present results demonstrated that PPS from the non-conventional but edible herb *P. oleracea* had a nutraceutical function in the colon, reflected by its anti-cancer activities in HCT-116 cells. A chemical selenylation of PPS using HNO_3_-Na_2_SeO_3_ could covalently conjugate the Se element into the PPS molecules, resulting in higher contents of Se for these two selenylated products, namely SePPS1 and SePPS2. Moreover, compared with PPS, this selenylation caused a change in the monosaccharide composition for SePPS1 and SePPS2, including reduced arabinose, fucose, and ribose contents, enhanced galactose, glucose, fructose, mannose, rhamnose, galacturonic acid, and glucuronic acid contents, and a stable xylose content. However, SePPS1 and SePPS2 shared similar monosaccharide compositions. Additionally, this selenylation endowed SePPS1 and SePPS2 with more enhanced activities than PPS in the cells, leading to the higher growth inhibition, enhanced disruption of mitochondrial membrane potential, greater promotion of ROS formation, higher apoptosis induction, and increased capacity to regulate the apoptosis-related genes and proteins involved in the ROS-mediated mitochondrial pathway. The higher selenylation extent could confer the selenylated products with higher activity in the cells. It is thus highlighted that selenylation might be effective and applicable to enhance the nutraceutical effect of PPS in the colon, to bring about an increased health-promoting function. It is also suggested that whether this selenylation has other beneficial impacts on natural polysaccharides needs further investigation, and whether the cultivation of edible plants in the Se-rich soil might cause an increase in the nutraceutical effects for natural polysaccharides are also interesting topics.

## Figures and Tables

**Figure 1 foods-11-03748-f001:**
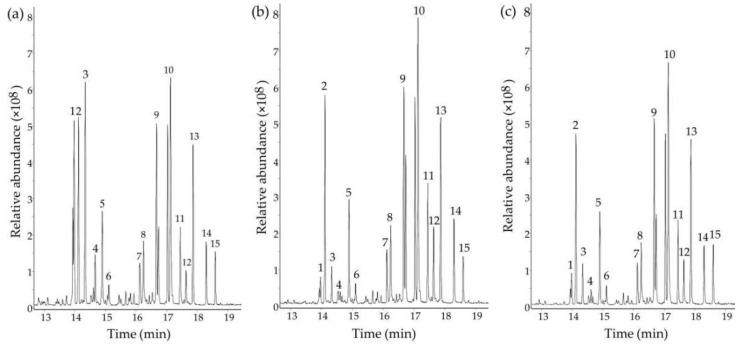
GC chromatograms for the hydrolyzed and derivated products from *P. oleracea* polysaccharides (PPS) with (**a**) Two selenylated PPS products SePPS1 and (**b**) SePPS2 (**c**). Peaks 1–4, arabinose, rhamnose, ribose, and fucose; peak 5, unidentified component; peaks 6–10, xylose, mannose, fructose, galactose, and glucose; peaks 11–12, unidentified components; peaks 13–14, galacturonic and glucuronic acids; peak 15, unidentified component.

**Figure 2 foods-11-03748-f002:**
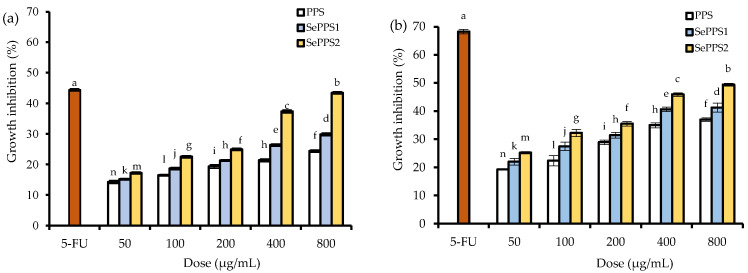
Proliferation inhibition (%) of positive control 5-Fu, *P. oleracea* polysaccharides (PPS), and two selenylated PPS products (SePPS1 and SePPS2) at the five dose levels on HCT-116 cells at 24 (**a**) and 48 h (**b**). Different lowercase letters above the column indicated significant differences of the mean values (*p* < 0.05).

**Figure 3 foods-11-03748-f003:**
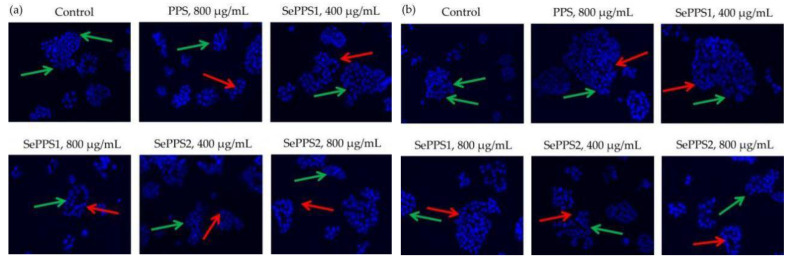
Hoechst 33258 staining results for HCT-116 cells treated with *P. oleracea* polysaccharides (PPS) and two selenylated PPS products (SePPS1 and SePPS2) at two dose levels for 24 (**a**) and 48 h (**b**), respectively. The green arrows indicate the normal cells, while the red ones indicate the apoptotic cells.

**Figure 4 foods-11-03748-f004:**
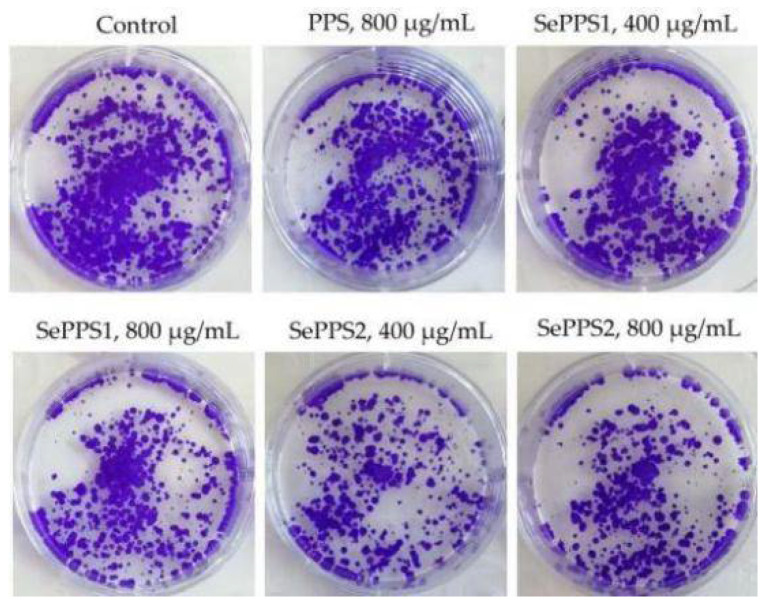
The observed inhibitory effects of *P. oleracea* polysaccharides (PPS) and two selenylated PPS products (SePPS1 and SePPS2) at two dose levels on HCT-116 cells with a treatment time of 14 d using the colony formation experiment.

**Figure 5 foods-11-03748-f005:**
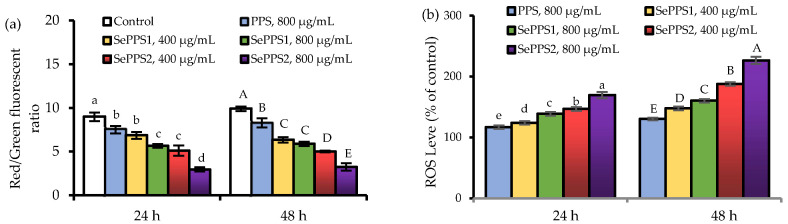
The measured ratios of red/green fluorescence (**a**) and ROS levels (**b**) for HCT-116 cells exposed to *P. oleracea* polysaccharides (PPS) and two selenylated PPS products (SePPS1 and SePPS2) for 24 and 48 h, respectively. Different lowercase or uppercase letters above the column indicated significant differences of the mean values (*p* < 0.05).

**Figure 6 foods-11-03748-f006:**
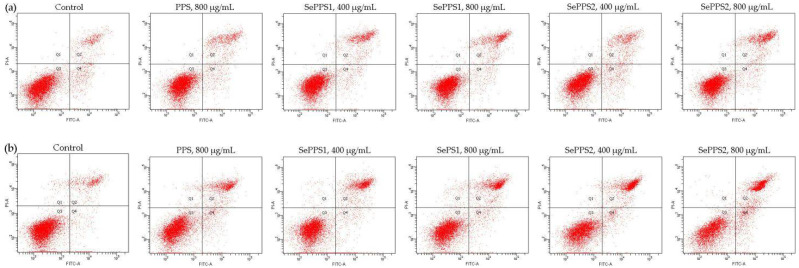
The apoptosis induction of *P. oleracea* polysaccharides (PPS) and two selenylated PPS products (SePPS1 and SePPS2) at two dose levels in HCT-116 cells with treatment times of 24 (**a**) and 48 h (**b**).

**Figure 7 foods-11-03748-f007:**
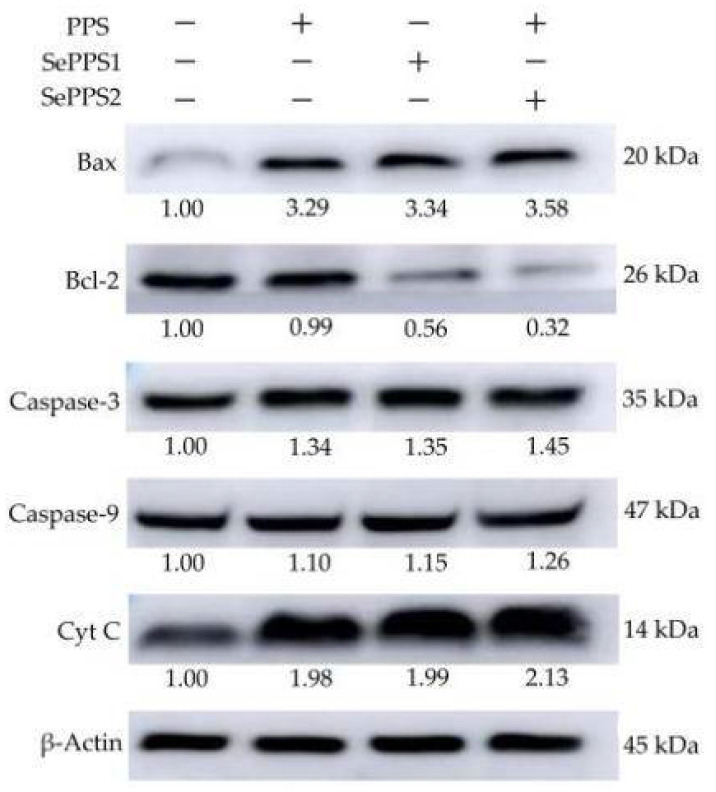
Effects of *P. oleracea* polysaccharides (PPS) and two selenylated PPS products (SePPS1 and SePPS2) on the expression of five protein targets in HCT-116 cells with a treatment time of 24 h. Cyt C, cytochrome C. The labeled values indicate expression changes in the targeted proteins.

**Table 1 foods-11-03748-t001:** Primer sequences used in the real-time PCR assay.

Gene		Sequence (5′-3′)
Bax	Forward Reverse	CAGTTTGCTGGCAAAGTAGAAA CGAACTGGACAGTAACATGGAG
Bcl-2	Forward Reverse	GACTTCGCCGAGATGTCCAG GAACTCAAAGAAGGCCACAATC
Caspase-3	Forward Reverse	CTGAATGTTTCCCTGAGGTTTG CCAAAGATCATACATGGAAGCG
Caspase-9	Forward Reverse	CTGCTGCGTGGTGGTCATTCTC TCGACCGACACAGGGCATCC
Cytochrome C	Forward Reverse	GCCAATAAGAACAAAGGCATCA TTAAGTCTGCCCTTTCTTCCTT
β-Actin	Forward Reverse	AACACCCCAGCCATGTACG ATGTCACGCACGATTTCCC

**Table 2 foods-11-03748-t002:** Relative average percentages of eleven monosaccharides detected in *P. oleracea polysaccharides* (PPS) and two selenylated PPS products (SePPS1 and SePPS2).

Monosaccharide	PPS (%)	SePPS1 (%)	SePPS2 (%)
Arabinose	12.69	2.04	2.60
Galactose	12.33	17.29	17.00
Fructose	2.46	4.44	4.62
Fucose	2.16	0.86	0.63
Galacturonic acid	7.56	9.29	9.30
Glucose	21.92	27.77	28.39
Glucuronic acid	2.88	4.46	4.19
Mannose	2.46	2.92	2.67
Ribose	10.13	1.62	2.29
Rhamnose	9.06	10.73	10.04
Xylose	1.05	1.13	1.09

**Table 3 foods-11-03748-t003:** The measured total apoptotic percentages (Q2 + Q4) of HCT-116 cells exposed to *P. oleracea* polysaccharides (PPS) and two selenylated PPS products (SePPS1 and SePPS2) for 24 and 48 h.

Cell Group	Dose Level (μg/mL)	Total Apoptotic Cells (%)	
		24 h	48 h
Control	None	10.7 ± 0.2	9.5 ± 0.3
PPS	800	14.8 ± 0.8	20.2 ± 0.5
SePPS1	400	16.8 ± 0.4	23.3 ± 0.3
	800	17.2 ± 0.2	24.3 ± 0.4
SePPS2	400	18.1 ± 0.5	31.9 ± 0.7
	800	20.5 ± 1.1	35.9 ± 0.9

**Table 4 foods-11-03748-t004:** Relative gene expression folds of HCT-116 cells treated with PPS, SePPS1, and SePPS2 for 24 h.

Gene	Polysaccharide Sample and Used Dose Level (μg/mL)				
	PPS, 800	SePPS1, 400	SePPS1, 800	SePPS2, 400	SePPS2, 800
Bax	1.28 ± 0.02	1.58 ± 0.03	1.85 ± 0.04	1.89 ± 0.05	2.30 ± 0.01
Bcl-2	0.96 ± 0.02	0.94 ± 0.03	0.82 ± 0.01	0.67 ± 0.03	0.61 ±0.01
Caspase-3	1.37 ± 0.02	1.47 ± 0.03	1.75 ± 0.02	2.35 ± 0.03	2.39 ± 0.04
Caspase-9	1.06 ± 0.02	1.18 ± 0.02	1.65 ± 0.01	1.78 ± 0.03	2.40 ± 0.03
Cytochrome C	1.04 ± 0.01	1.28 ± 0.02	1.36 ± 0.04	1.57 ± 0.02	1.64 ± 0.05

PPS, *P. oleracea* polysaccharides; SePPS1 and SePPS2, selenylated PPS products with Se contents of 840.3 and 1770.5 mg/kg, respectively.

## Data Availability

All data are contained within the article.

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
