# Peer review of "Monosaccharide Composition and In Vitro Activity to HCT-116 Cells of Purslane Polysaccharides after a Covalent Chemical Selenylation"

_foods, 2022, doi:10.3390/foods11233748_

Round 1
Reviewer 1 Report
The manuscript is well explained and designed. the authors discussed the topic how chemical selenylation of PPS can alter the saccharide composition and nutraceutical effect of the human colon cancer HCT-116 cells. The topic being discussed is also of great significance and importance. I have a few thoughts/modifications that I would like to suggest:
Abstract: need an opening statement to introduce the topic. the abstract currently reads as the results section, this section needs to be re-worked on to highlight the important results that tests the hypothesis. A conclusion statement about the overall significance of the work is also required. Where does this work fits in the overall picture (how did you address or aimed to address the gaps in literature).
Introduction: in the last paragraph instead of talking about methods, rather use the results from those different experiments. The results will need to be highlighted rather than the methods as well as a conclusion of the work.
Methods:
1. section 2.2: how long were the cells digested for, you mention that it was done every 23 d
2. section 2.3: line 132 'incubated' instead of 'stood'
line 153: 'quickly' is not a scientific term
line 176: photographed? how and the images were they processed?
line 195: cells were then imaged using microscope ..
Overall the methods section is too long and there is alot of repetition between the different sections. different sections can be incorporated together to avoid repetition of methods highlighting the differences for each experiment that was run.
Results section:
Figure 1: make the chromatograms larger.
Figure 2: make the figures larger and also improve the resolution of the figure (some of the text is not visible)
Figure 3: i would suggest to use a different color for one of the arrows (i.e. red or green) might be hard to differentiate between the two
table 4: legend is not on the same page as the table
Overall the results section must be improved by incorporating statistics into the text when comparing treatments and showing either an increase or decrease for all the sections. The figures can also be improved by improving the overall resolution of them and making some of the figures larger for ease to read.
Discussion:
Rather than backing the literature with your results. You should discuss the results in light of the literature. Currently reads as if nothing new has been published and it seems redundant to repeat what was done previously. What is the highlight of the text in the first paragraph, how does it address gaps in the literature. it seems that the literature (what is already known) is more discussed than the actual results and what they mean in the context of this manuscript.
The Discussion section still requires more work.
Author Response
The manuscript is well explained and designed. the authors discussed the topic how chemical selenylation of PPS can alter the saccharide composition and nutraceutical effect of the human colon cancer HCT-116 cells. The topic being discussed is also of great significance and importance. I have a few thoughts/modifications that I would like to suggest:
Reply: The authors thank the reviewer for her/his kind comment.
Abstract: need an opening statement to introduce the topic. the abstract currently reads as the results section, this section needs to be re-worked on to highlight the important results that tests the hypothesis. A conclusion statement about the overall significance of the work is also required. Where does this work fits in the overall picture (how did you address or aimed to address the gaps in literature).
Reply: Thanks for your suggestion.
Based on your suggestion, we have added a topic introduction in the abstract section. Please see the revisions in the lines 18-20.
Introduction: in the last paragraph instead of talking about methods, rather use the results from those different experiments. The results will need to be highlighted rather than the methods as well as a conclusion of the work.
Reply: A good suggestion.
We have rewritten the last sentence as “This study aimed to reveal whether monosaccharide composition and nutraceutical effect (to be specific, anti-cancer effect) of PPS could be impacted by the performed chemical selenylation as well as the obtained selenylation extent.”
Thanks!
Methods:
- section 2.2: how long were the cells digested for, you mention that it was done every 23 d
Reply: Sorry, it was a pen slip.
The cells were digested with 0.25% trypsin digestion solution every 2-3 days for 1 minute. Please see the revisions in the line 133.
- section 2.3: line 132 'incubated' instead of 'stood'
Reply: Thanks! The related word“stood” has been replaced by“incubated”, based on your suggestion
We checked out the whole manuscript, and made corrections accordingly.
line 153: 'quickly' is not a scientific term
Reply: Sorry for the unsuitable word use.
We have replaced 'quickly' with 'immediately'. Please see the revision in the line 167. Thanks!
line 176: photographed? how and the images were they processed?
Reply: Thanks for your valuable comment.
These images were obtained using a microscope, not processed, and given in the manuscript as the original picture.
line 195: cells were then imaged using microscope ..
Reply: Thanks for your comment.
The sentence has been modified to the cells that were then imaged using a microscope (Zeiss Axio Observer A1m, Carl Zeiss, Oberkochen, BW, Germany) with an objective of 20-fold. Please see the revisions in the lines 212-214.
Overall the methods section is too long and there is a lot of repetition between the different sections. different sections can be incorporated together to avoid repetition of methods highlighting the differences for each experiment that was run.
Reply: Thanks for your comment.
In the method section, I would like to give a detailed experimental procedure for the convenience of the readers.
Results section:
Figure 1: make the chromatograms larger.
Reply: Thanks!
To control publishing space, we prepared the pictures in acceptable sizes. The editor might modify the picture size during paper conversion.
Figure 2: make the figures larger and also improve the resolution of the figure (some of the text is not visible)
Reply: Thanks! The editor might give a final selection or revision.
Figure 3: i would suggest to use a different color for one of the arrows (i.e. red or green) might be hard to differentiate between the two
Reply: Thanks for your comment.
For Figure 3, we have used two different colored arrows; i.e. the green for normal cells while the red for concentrated chromatin or apoptotic bodies.
table 4: legend is not on the same page as the table
Reply: Thanks for your comment.
We have revised the table.
Overall the results section must be improved by incorporating statistics into the text when comparing treatments and showing either an increase or decrease for all the sections. The figures can also be improved by improving the overall resolution of them and making some of the figures larger for ease to read.
Reply: Thanks for your comment.
The whole Results section has been rewritten. Please see these words marked with red. In addition, we added the analysis methods of total saccharide and ash. Please see the revisions the lines 157-159.
Discussion:
Rather than backing the literature with your results. You should discuss the results in light of the literature. Currently reads as if nothing new has been published and it seems redundant to repeat what was done previously. What is the highlight of the text in the first paragraph, how does it address gaps in the literature. it seems that the literature (what is already known) is more discussed than the actual results and what they mean in the context of this manuscript.
Reply: Thanks for these valuable suggestions.
In the first paragraph of the discussion, we show that selenylated polysaccharides have anti-cancer potential through literature, and this potential is shown in inhibiting cell proliferation or inducing cell apoptosis. Therefore, by evaluating the indicators of cell proliferation and apoptosis, we can provide a conclusion that these selenylated P. oleracea polysaccharides had an anti-cancer role by inhibiting cell proliferation and inducing cell apoptosis, and were more active than the original PPS.
The Discussion section still requires more work.
Reply: Thanks!
In this section, we show three facts: (1) the selenylated polysaccharides should have enhanced activity; (2) how to verify the selenylated polysaccharides having these activities; and (3) the related pathway.
In our personal opinion, these discussed topics are acceptable.
Thanks again!
Reviewer 2 Report
The manuscript entitled " Monosaccharide Composition and in Vitro Activity to HCT-116 Cells of the Purslane Polysaccharides as Affected by a Covalent Chemical Selenylation" evaluated Portulaca oleracea polysaccharides chemically selenylated using the HNO3-Na2SeO3 method and evaluated their effects on Growth inhibition of HCT-116 cells, apoptosis and ROS formation. It is very interesting and useful scientific work which cause an increase in nutraceutical effect for this edible plant and I have some comments:
1. In the 2.1. Materials and Reagents section, kindly add when the fresh plant was collected, from where, who identified it and its voucher specimen code.
2. Throughout the whole manuscript, replace the Purslane word with P. oleracea.
3. In the 2.3. Preparation and Chemical Selenylation of Purslane Polysaccharides section explain the used drying procedure you followed, as it's important for plants constituents.
4. In the material and method section kindly add the parts used from P. oleracea plant in your study, e.g. leaves, flowers, stalks.
5. The manuscript needs major grammar, typos, and editing correction by a native speaker
Author Response
The manuscript entitled "Monosaccharide Composition and in Vitro Activity to HCT-116 Cells of the Purslane Polysaccharides as Affected by a Covalent Chemical Selenylation" evaluated Portulaca oleracea polysaccharides chemically selenylated using the HNO3-Na2SeO3 method and evaluated their effects on Growth inhibition of HCT-116 cells, apoptosis and ROS formation. It is very interesting and useful scientific work which cause an increase in nutraceutical effect for this edible plant and I have some comments:
- In the 2.1. Materials and Reagents section, kindly add when the fresh plant was collected, from where, who identified it and its voucher specimen code.
Reply: A good suggestion. Prof. Che had identified this raw material.
Please see the added information in the lines 113-115. Thanks!
- Throughout the whole manuscript, replace the Purslane word with P. oleracea.
Reply: Thanks for your comment.
The word “Purslane” has been replaced by “P. oleracea” throughout the whole manuscript, according to your suggestion.
- In the 2.3. Preparation and Chemical Selenylation of Purslane Polysaccharides section explain the used drying procedure you followed, as it's important for plants constituents.
Reply: Thanks for your comment.
We have added the drying process of purslane. Fresh purslane was dried in the shade. Please see the revisions in the lines 136-138.
Fresh Purslane dried in the shade was chosen as experimental material
- In the material and method section kindly add the parts used from P. oleracea plant in your study, e.g. leaves, flowers, stalks.
Reply: Yes. Please see the revisions in the lines 137-138. Thanks!
- The manuscript needs major grammar, typos, and editing correction by a native speaker
Reply: Thanks for your valuable suggestion.
The whole manuscript was rewritten, aiming to improve its writing quality. Another researcher who made contribution for this paper revision was thus listed as one of the co-authors.
Thanks again!
Round 2
Reviewer 1 Report
I thank the authors for considering my previous comments.
I would recommend a proof-reading check prior to final acceptance.
Reviewer 2 Report
The authors conducted all required corrections. Thank you